# Design of Alkali-Activated Materials and Geopolymer for Deep Soilmixing: Interactions with Model Soils

**DOI:** 10.3390/ma17153783

**Published:** 2024-08-01

**Authors:** Faten Souayfan, Emmanuel Roziere, Michael Paris, Dimitri Deneele, Ahmed Loukili, Christophe Justino

**Affiliations:** 1École Centrale Nantes, Nantes Université, CNRS, GeM, UMR 6183, F-44000 Nantes, France; 2Soletanche-Bachy, Chemin des Processions, F-77130 Montereau Fault Yonne, France; 3Nantes Université, CNRS, Institut des Matériaux de Nantes Jean Rouxel, IMN, F-44000 Nantes, France; 4GERS-LGIE, Université Gustave Eiffel, F-44344 Bouguenais, France

**Keywords:** soilmixing, geopolymer, alkali-activated materials, grout, NMR, compressive strength

## Abstract

This study focuses on the use of alkali-activated materials and geopolymer grouts in deep soilmixing. Three types of grouts, incorporating metakaolin and/or slag and activated with sodium silicate solution, were characterized at different scales to understand the development of their local structure and macroscopic properties. The performance of the soilmix was assessed by using combinations of the grouts and model soils with different clay contents. Feret’s approach was used to understand the development of compressive strength at different water-to-solid ratios ranging from 0.65 to 1. The results suggested that incorporating calcium reduced the water sensitivity of the materials, which is crucial in soilmixing. Adding soils to grouts resulted in improved mechanical properties, due to the influence of the granular skeleton. Based on strength results, binary soilmix mixtures containing 75% of metakaolin and 25% of slag, with H_2_O/Na_2_O ratios ranging from 28 to 42 demonstrated potential use for soilmixing due to the synergistic reactivity of metakaolin and slag. The optimization of compositions is necessary for achieving the desired properties of soil mixtures with higher H_2_O/Na_2_O ratios.

## 1. Introduction

Constructing buildings and infrastructure on clayey soils, soft soils, and soils with high permeability presents a significant challenge. Deep soil mixing has emerged as a technique to improve soil properties and has gained increasing popularity in recent years. This technology improves the physical and mechanical properties of soil by mixing it in situ with a diluted cement paste called grout, using specific tools [1]. The grouts often rely on Portland cement due to its advantageous mechanical strength, availability, and cost-effectiveness. However, the production of Portland cement results in significant CO_2_ emissions and high energy demands [2,3,4], and the durability of cement-based materials is questioned when subjected to aggressive waters [5,6]. In this context, the soil improvement industry is seeking alternative binders to Portland cement to address environmental concerns and enhance durability.

Alkali-activated materials, produced by mixing solid aluminosilicate sources such as fly ash, metakaolin, and slag with an alkaline solution, suggest new opportunities for grouting and soil enhancement. Numerous studies have examined the use of high-calcium alkali-activated materials, such as ground granulated blast furnace slag and high-calcium fly ash, as soil stabilizers [7,8,9,10]. Low and high-calcium fly ash-based materials were assessed for their effectiveness in improving deep soft soil. Their research involved alkali-activated fly ash grouts mixed with soft soils [11,12]. The performance of fly ash-based materials was comparable to that of cement and lime for stabilizing deep soft soils. Other studies have shown the potential of low-calcium binders as soil stabilizers [12,13,14]. The reactive material dissolves to form aluminosilicate gel, which binds the solid particles together, resulting in a more uniform and compact microstructure after stabilization [13]. Significant improvements in strength and stiffness were observed when sandy loam soil was mixed with class F fly ash, sodium silicate, and sodium hydroxide solution [15]. Previous studies [8,10] have examined the use of combined fly ash and slag geopolymers for stabilizing soft clay at various activator solution-to-solid ratios. The mechanical strength increased with the activator-to-solid ratio up to a certain limit, after which early precipitation hindered the formation of an effective geopolymer network. Moreover, it was found that the strength of silty clay soil treated with fly ash was affected by several factors, including the Na_2_SiO_3_/NaOH ratio, the activator-to-solid ratio, and temperature conditions [16].

Research on soil improvement using metakaolin-based alkali-activated materials has remained limited. At present, metakaolin seems to meet various economic, performance, and environmental criteria [15]. It has the advantage of being produced from natural clayey materials with a relatively low carbon footprint [16,17] and leading to relatively good durability [18,19,20]. The potential of metakaolin-based geopolymers to stabilize sulfate-rich soils was demonstrated [13,14]. However, additional research is needed to investigate the use of alkali-activated grout mixtures in deep soil mixing techniques. Several challenges related to the development of these grouts need to be addressed.

Grout mixtures must meet various specifications in their fresh and hardened states. Fresh grouts should be stable during storage and before injection, fluid enough to form a homogeneous soil–grout mixture, and maintain these properties throughout the soil mixing process [21,22,23]. Mechanical properties and durability are essential to ensure short-term and long-term soil stabilization. As mechanical compaction is not used in soil mixing, grouts require a high initial liquid fraction to ensure self-compaction during injection. When soil is mixed with the grout, the water from the soil increases the total water-to-solid ratio of the mixture (see Figure 1). The amount of water added varies with the type of soil and its moisture content. Therefore, it is crucial to monitor the liquid fraction, as it has a significant impact on the mechanical properties of the soil–grout mixtures.

Various studies have highlighted that water content is a key parameter in the synthesis of geopolymers and alkali-activated materials for the development of mechanical strength [24,25]. Recent research on metakaolin-based geopolymers has shown a linear decrease in mechanical strength as water content increases [26]. Another study revealed that the strength and microstructure of metakaolin-based grouts are significantly affected by water content compared to slag-based grouts [27]. A higher water-to-solid ratio affects the reactivity of metakaolin-based grout, resulting in a decrease in compressive strength [28]. However, low-calcium binders are less vulnerable to degradation in aggressive environments. The development of binary binders combining metakaolin and slag in alkali-activated materials is promising. The properties of these binders were improved compared to plain activated metakaolin or slag [29,30,31]. A meta-analysis was developed to characterize their phase composition by analyzing NMR data [32].

This paper aims to explore the use of new materials for soil improvement through the deep soil mixing technique. While existing research on alkali-activated materials primarily addressed concentrated mixtures for concrete technology, this study focused on developing and characterizing three types of diluted mixtures: metakaolin, slag, and metakaolin–slag-based grouts activated with a sodium silicate solution. The grout compositions were designed to replicate the phenomena observed in soil mixing at the grout scale. The physico-chemical properties throughout the structuration process were monitored from the initial stages to the hardened state. To determine whether the grouts meet the specified requirements, it is crucial to consider both the type of soil and the water content. Reproducing real soil conditions is challenging due to the effects of various factors, such as complex clay composition and organic matter. Therefore, model soils were mixed with the grouts to create soil mixtures, and the water sensitivity of different grouts was evaluated and compared with soil mixtures of the same H_2_O/Na_2_O ratios. The influence of the water-to-solid ratio on compressive strength was analyzed as a function of binder composition.

## 2. Materials and Methods

### 2.1. Materials

The alkali-activated materials in this study were prepared from dry powders and aqueous solutions with known chemical compositions. A sodium silicate solution with a fixed SiO_2_/Na_2_O ratio of 1.7 and 44% dry content in demineralized water was used. This solution was formulated to have a relatively high Na_2_O content, which minimizes the subsequent addition of NaOH. A small amount of NaOH pellets was added to adjust the Na/Al molar ratio. The SiO_2_/Na_2_O molar ratio of the activator and the chemical compositions were chosen based on literature recommendations to ensure adequate dissolution of the precursors [33].

The dry powders (precursors) used in the study were metakaolin from Fumel, France, and ground granulated blast furnace slag (GGBFS) from Fos, France. Metakaolin mainly consists of silica (67.1 wt.%) and alumina (26.8 wt.%). The high silica content is attributed to the significant amount of quartz; its proportion of pure metakaolin Al_2_O_3_(SiO_2_)_2_ phase was 43%. [15]. The ground granulated blast furnace slag (GGBFS) contained a significant amount of CaO. The quantitative chemical compositions and physical properties of the precursors are presented in Table 1. Their particle size distribution and X-ray diffraction patterns are presented in Figure 2. The X-ray diffraction pattern shows that the studied slag predominantly included amorphous phases, as indicated by the presence of a broad hump in the 20–38° 2Ө range. The major crystalline phase in raw metakaolin was quartz (SiO_2_).

### 2.2. Testing Program

This study investigates the use of alkali-activated materials as diluted mixtures (grouts) in soil mixing. Three types of grouts were studied: The binary metakaolin–slag grout MKS28 and the plain slag grout S28 were derived from the 100% metakaolin grout MK28. In these mixtures, metakaolin was substituted with 25% slag for MKS28 and 100% slag for S28, while maintaining constant volume. These proportions enable a trade-off between the rheological, mechanical, and durability properties of slag-metakaolin-based materials [32]. Low bentonite content was added for the stabilization of fresh grouts. The composition of these grouts is presented in Table 2.

The grouts were designed with an H_2_O/Na_2_O molar ratio of 28, which allows the analysis of a relatively diluted system while maintaining significant mechanical strength (Table 2). The water-to-solid ratio was calculated based on the total water in the mixture, which includes both the water from the activation solution and the added demineralized water. The total solid includes both the precursor and the solid fraction of the activation solution. The Si/Al and Na/Al ratios of the grouts were determined by considering the amorphous component of metakaolin and the overall composition of slag.

In the second phase of the study, the mechanical performance of grouts was compared and assessed at various H_2_O/Na_2_O ratios to understand the effect of water in deep soil mixing at the grout level (see Table 3). The mixtures were prepared with different water-to-solid ratios and a constant solid-to-sodium silicate solution mass ratio. As a result, the mixtures present constant Si/Al and Na/Al molar ratios and different H_2_O/Na_2_O ratios. For this part of the study, diluted slag mixtures were used as the S28 mixture was relatively concentrated in silicates.

Model soils were created using silica powder (S100) and a blend of silica powder with Speswhite kaolinite (K50S50). The mass proportions of silica powder and kaolinite were 50%. Speswhite kaolin (K) has a liquid limit (WL) of 55% and a plasticity index (PI) of 25%, with an initial water content of 40%, which falls between its liquid limit and plastic limit. Plain metakaolin and binary metakaolin–slag grouts, with an initial H_2_O/Na_2_O ratio of 21 (MK21 and MKS21), were mixed with the model soils. The total water content of the soil mix (soil + grout) was adjusted to achieve H_2_O/Na_2_O ratios of 28, 34, and 42 (in the case of binary mixtures). Grout was added to the soil at a volume fraction of 50%. The mechanical strength of these mixtures was assessed to evaluate the impact of water content, precursors, and soil type. Table 4 provides an example of the composition for 2 L of soilmix incorporating 1 L of plain metakaolin grout MK21 and 1 L of model soil, and the resulting ratios for the grouts in soilmix. The grout mass in the soilmix includes the mass of MK21 grout and the water content of the soil.

### 2.3. Sample Preparations and Testing Methods

The activation solution was prepared, and then the powders were added and stirred in a high-shear mixer for 5 min to ensure proper homogenization. Subsequently, the grouts were analyzed using the experimental procedures outlined below. For the soil mix preparation, the soil was initially prepared, and water was added to reach the desired water content. If the soil included kaolin, it was mixed with water at 40% content 24 h prior to sample preparation and stored at 20 °C. The grout was then combined with the soil using a Hobart mixer (Hobart, Troy, OH, USA)until completely homogenized. Specimens were sealed and stored at 20 °C for characterization in their hardened state.

#### 2.3.1. Nuclear Magnetic Resonance

To conduct solid-state NMR spectroscopy, a Bruker NEO 300 MHz (7 T) spectrometer (Bruker, Billerica, MA, USA) and a 7 mm MAS probe were used for ^29^Si NMR, while a 500 MHz Bruker Avance III with a 2.5 mm MAS probe was used for ^27^Al magic angle spinning (MAS) NMR. For ^27^Al acquisition, the MAS speed was 30 kHz and the repetition time was set to 1 s. A single π/13 excitation pulse was used for a radiofrequency field of 12 kHz. Chemical shifts were referenced against Al(NO_3_)_3_ solution. For ^29^Si acquisition, the MAS speed was 5 kHz, and the 29Si MAS NMR spectra were acquired with a single π/2 excitation of 5.5 μs. The recycle delay between scans was set to 5 s. Chemical shifts were referenced against TMS.

#### 2.3.2. Compressive Strength

The compressive strength testing was carried out on a 100 kN press. The load rate was 1.9 kN/s until failure [34]. Compression tests were performed at the ages of 7, 28 and 90 days on cylinders with a diameter of 7 cm and a height of 14 cm according to the NF EN 12390-1 standards [35].

#### 2.3.3. Isothermal Calorimetry

The short-term reactivity of studied mixtures was characterized by using an isothermal calorimeter of the TAM Air type (TA Instruments, New Castle, DE, USA) at 20 °C. The heat flow was measured, and the cumulative heat was deduced by the integration of the measured flux. The measurements were carried out for 7 days after the end of mixing.

## 3. Results

### 3.1. Grout Characterization

#### 3.1.1. Long-Term Local Structure

^27^Al NMR spectra of the activated grouts and raw metakaolin after 250 days are presented in Figure 3 The raw metakaolin displays three overlapping peaks at 55 ppm, 28 ppm, and 4 ppm, corresponding to four, five, and six-coordinated aluminum, respectively. The broad peaks are a result of the highly disrupted geometry of all three aluminum sites [33]. The slag spectrum reveals broad overlapping peaks associated with different aluminum environments, predominantly Al (IV).

The ^27^Al NMR spectrum of the activated slag shows visible peaks at 74 ppm and 68 ppm, both corresponding to Al (IV) (q2 sites) in C-A-S-H [36,37]. The aluminate peak at 9 ppm is attributed to Al (VI) in hydrotalcite within a secondary product of slag alkali-activation due to the non-negligible MgO content of slag. Hydrotalcite is considered a poorly crystallized phase, and it is not always detectable by XRD [38,39].

The activated metakaolin grout MK28 and the binary metakaolin–slag mixture MKS28 display a primary resonance peak with a maximum between 58 and 60 ppm. This peak is associated with tetrahedrally coordinated AlO_4_(4Si) within a three-dimensional silico-aluminate framework [40,41,42]. Furthermore, spectral intensity below 40 ppm, corresponding to Al (V) and Al (VI) signals, suggests that some of the metakaolin remained unreacted even after 250 days.

^29^Si NMR spectrum of raw metakaolin shows a broad and asymmetric peak indicating the amorphous nature of the material (Figure 4). The peak at −107 ppm is attributed to the presence of quartz in raw metakaolin [37]. The spectrum of raw slag lies in a variety of Q^0^, Q^1^, and Q^2^ sites, mostly Q^0^ resulting in a peak maximum of around −75 ppm. The ^29^Si NMR spectrum of activated slag shows well-defined peaks at chemical shifts of −79 ppm, −82 ppm, and −85 ppm, corresponding respectively to Q^1^, Q^2^(1Al), and Q^2^ sites in the C-A-S-H structure [43,44,45,46].

The activated metakaolin system MK28 shows a broad resonance peak ranging from −91 ppm to −93 ppm, which is attributed to Q^4^(mAl) type environments with values of m varying from 0 to 4 [37]. Comparing MKS28 to plain metakaolin mixtures reveals a shift in the spectrum towards higher frequencies, ranging from −72 ppm to −104 ppm. The phase composition of these binary mixtures has been previously discussed and analyzed in our research [32]. The results suggested the formation of a heterogeneous phase involved in the transformation of the 3D network to C-A-S-H. Consequently, the different phases present in plain metakaolin, plain slag, and binary mixtures may influence grout behavior on both micro and macroscopic levels.

#### 3.1.2. Reaction Advancement

The reaction rates and heat flow profiles of the studied grouts differ significantly (Figure 5). The slag grout S28 shows a pronounced peak followed by a ‘dormant’ period lasting several hours before the main heat release begins. The metakaolin grout MK28 mixture undergoes two distinct stages of dissolution and polycondensation [47].

The binary mixture MKS28 exhibits a single prominent exothermic peak with a shoulder during the initial hours of the reaction. Following this peak, the heat flux gradually decreases over time, consistent with previous studies [33]. During this period, silicates, aluminates, and calcium units form simultaneously. The dissolution of metakaolin and slag species depends on the activation conditions.

The heat flow intensity varies significantly, with the MK28 and MKS28 mixtures showing lower heat flow compared to S28. This difference is likely due to the different reactivity of metakaolin and slag, with the substantial CaO content in slag playing a role.

The heat flow for the hybrid mixture MKS28, shown by the dashed line in Figure 5 (“weighted sum”) was computed as a linear combination, incorporating 75% of the heat flow from MK28 and 25% from S28.
(1)X MKS28=0.75×X MK28+0.25×X S28
where X represents a specific property, in this case, the heat flow.

The resulting curve does not correspond to the measured heat flow profile of MKS28, suggesting a synergistic interaction between metakaolin and slag during the first 20 h. The addition of calcium to metakaolin systems enhances metakaolin dissolution and increases the extent of the reaction [48,49,50]. In a calcium-rich medium, the formation of C-A-S-H at an early stage of the reaction consumes silicate from the interstitial solution, which promotes the dissolution of metakaolin to maintain the concentration of silicate ions [51]. Chen et al. [52] mentioned that the addition of calcium reduces the possibility of the formation of an aluminosilicate gel layer on the surface of sodium silicate-activated metakaolin.

#### 3.1.3. Microstructure and Mechanical Properties

The chemically bound water is calculated from Equation (2) and plotted as a function of compressive strength at 28 and 90 days (Figure 3).
(2)$$\%\;chemically\;bound\;water=\frac{\left(Mdry-MdryT\right)}{MdryT}\times 100$$

Mdry and MdryT correspond to the dry mass of the specimen stored in an oven at 105 °C and the mass of dry materials in the specimen deduced from the mix design (amount of metakaolin and/or slag and the dry fraction of sodium silicate solution), respectively. The strength is directly proportional to the amount of bound water (Figure 6). The low bound water content in the geopolymer formed in the activated metakaolin systems accounts for the significant difference in strength between the mixtures. Given that the mixtures had the same initial water volume (Table 2) and therefore the same initial porosity, lower bound water content implies higher porosity.

The binary grout MKS28 exhibits a behavior that is intermediate between plain metakaolin and slag grouts. It is important to note that the amount of bound water in MKS28 was not directly proportional to the slag content, as indicated by the values obtained from the weighted sum of bound water in MK28 and S28 (Equation (1))—MKS28 had a higher bound water content. This may be attributed to the nature of the products formed, as discussed in our earlier study on the microstructural characterization of binary metakaolin–slag mixtures [32].

### 3.2. Effect of H_2_O/Na_2_O on Mechanical Properties: Grout Scale

In this section, the soil mixing process was simulated at the grout level. The water-to-solid ratio and the H_2_O/Na_2_O ratio were adjusted to account for the additional water from the soil (see Table 3). Compressive strength (fc) was assessed after 28 days.

The water-to-solid ratio is expected to influence strength at two levels: the properties of the binding phase and the initial porosity. Initially, it can be assumed that the properties of the binding phase are not affected and porosity only accounts for the variations of strength. This uncoupled approach corresponds to Feret’s equation, which is commonly used to predict the effect of initial composition on the compressive strength of cement-based materials (Equation (3)).
(3)$$fc=k\times \frac{C^{2}}{({C+W+A)}^{2}}$$
where k is a constant that depends on the properties of the aggregates and cement. C, W and A are the volumes of cement, water and air, respectively. Feret’s equation can be adapted to analyze the behavior of the studied alkali-activated materials considering:
C as the initial volume of reactive solids in the material (precursor and solid fraction of the activation solution).W as the total water content in the mix.A as the air volume is considered negligible in the material (confirmed by measurements of initial density).


The experimental compressive strength and the values calculated by Feret’s equation are plotted in Figure 7. The least squares method was used to calculate the constant k in Feret’s equation. The *x*-axis represents the initial porosity factor or initial packing density of the mixture $\frac{C^{2}}{{\left(C+W\right)}^{2}}$ (Equation (2)) where the volume of air is considered negligible (checked by the measurement of the initial density of fresh grouts).

The compressive strength of alkali-activated metakaolin-based grouts is not proportional to the initial packing density factor (see Figure 7a). It decreases exponentially as the initial packing density decreases. This suggests that higher water content may affect the reactivity of the precursor, leading to a lower proportion of the binding phase and/or changes in the intrinsic properties of the reaction products. NMR spectral analyses have indicated that an increased H_2_O/Na_2_O ratio in these grouts results in a higher amount of unreacted precursors and a lower proportion of reaction products without changing the local structure of products as reported in a previous study [28].

In plain slag mixtures (Figure 7b), the experimental results align with the values calculated using Feret’s equation, indicating that the reactivity of slag was not affected by the H_2_O/Na_2_O ratio. In activated slag mixtures, C-A-S-H phases are formed, and a significant amount of water is chemically bound (Figure 6). However, in metakaolin-based grouts, water remains in capillary pores. This difference in the role of water in the two systems influences the formation of reaction products and subsequently affects the physicochemical properties in the hardened state.

In metakaolin–slag mixtures, compressive strength decreases linearly with initial packing density, but it does not align with Feret’s values (Figure 7c). The reduction in compressive strength observed as the initial packing density decreased from 0.11 to 0.055 is more pronounced than what was predicted by Feret’s equation. In contrast to plain metakaolin, where ‘k’ decreases exponentially with the H_2_O/Na_2_O ratio, the incorporation of slag reduced the effect of water on the reactivity of the precursors and binder properties.

Based on these findings, the compressive strength of alkali-activated metakaolin and metakaolin–slag mixtures cannot be fully explained by the initial packing density within the H_2_O/Na_2_O range of 21 to 34. Adjustments to Feret’s equation are necessary to accurately represent the experimental compressive strength. The parameter ‘k’, which is generally considered constant for a given binder, appears to be influenced by the H_2_O/Na_2_O ratio. More data are needed to develop a modified Feret’s model suitable for alkali-activated materials in this H_2_O/Na_2_O range. In the case of alkali-activated slag, Feret’s approach indicates a behavior close to cement-based grouts within the H_2_O/Na_2_O range of 38 to 62. The concentration of the pore solution had minimal impact on the reactivity of the precursors, thus maintaining a constant value for ‘k’.

From these data, slag-based grouts appear more suitable for soil mixing compared to plain metakaolin and binary metakaolin–slag mixtures. However, the rapid hardening of these grouts could limit their use for deep soil mixing [53]. Additionally, high calcium binders are more vulnerable to some chemically aggressive waters [5], due to the dissolution of the hydration products. Therefore, the following section will focus on evaluating metakaolin and metakaolin–slag mixtures at the soil mixing scale, as these mixtures offer promising rheological properties and improved physico-chemical stability [18,54].

### 3.3. Deep Soilmixing Model Materials

This section examines the properties of soil and grout mixtures (soilmix) made by combining model soils with metakaolin-based grout (MK21, Table 3) and binary metakaolin–slag grout (MKS21, Table 3). While this study does not replicate the actual in-situ mixing process, it provides an initial laboratory investigation into the effects of factors such as soil characteristics and water content on the properties of soilmixes made with alkali-activated grouts.

#### 3.3.1. Influence of Soil on Reaction Advancement and Local Structure

The reaction advancement of plain metakaolin-based soilmixes was assessed using isothermal calorimetry (Figure 8). Table 4 details their compositions. The cumulative heat of soilmixes was reported per unit mass of grout and compared with the cumulative heat of plain grouts with H_2_O/Na_2_O ratios of 28 and 34 (represented by dashed lines), respectively. The grout mass in the soilmix includes the mass of MK21 and water content in the soil. The results show that the heat flow curves of the soilmixes align with those of the corresponding grouts, suggesting that the soil did not strongly participate in the reaction.

The impact of soil type on local structure was examined for plain metakaolin-based soilmixes at two H_2_O/Na_2_O ratios using ^29^Si and ^27^Al NMR spectra (Figure 9). The spectra of the soilmixes were compared with the grouts presented by the dotted lines in Figure 9.

In kaolin, a prominent peak at approximately −91 ppm is observed, corresponding to Q^3^ sites within the tetrahedral layers of kaolinite sheets, while a distinct peak at −107 ppm, associated with Q^4^ sites, is attributed to quartz. For crushed silica sand, a sharp peak at −107 ppm confirms its crystalline structure. The ^29^Si spectrum of the soilmix (S100 + MK21) with an H_2_O/Na_2_O ratio of 28, measured at 7 days, shows a range of peaks corresponding to various silicon species present in the mixture. The distributed peaks in the ^29^Si spectrum of the soilmix (S100 + MK21) are most likely Si species that have not yet reacted with the activation solution. However, after 28 days, a broad peak centered at −94 ppm appears, indicating the presence of Si in Q^4^(mAl) type environment. This spectrum shows a slight shift in the characteristic peak compared to the grout (MK28), suggesting that a higher amount of metakaolin did not react in the presence of ground silica sand compared to the grout alone. A similar observation is noted for the 100% crushed silica sand mixture, with a higher H_2_O/Na_2_O ratio. In the mixture of 50% kaolin and 50% crushed sand, a prominent peak at −91 ppm is observed, representing silicon in the tetrahedral layer of kaolinite, which predominates over other silicon species.

The ^27^Al spectra of kaolin display an aluminum peak at 3 ppm, indicating octahedral coordination. The ^27^Al spectra of soilmixes containing 100% ground silica sand align with those of the grouts. In mixtures comprising 50% kaolin and 50% ground silica sand, a peak at 4 ppm is observed, corresponding to Al (VI) in the octahedral layer of kaolinite. After 250 days, the peak at 58 ppm shows a significant increase, indicating that aluminum originally present in kaolinite or unreacted metakaolin is transforming into Al (IV) and integrating into the geopolymer network.

Based on these observations, silica sand may affect the advancement of the reaction, even if it does not participate directly. However, this suggestion may be questioned due to the minimal difference between the grouts and soilmixes spectra. Similarly, kaolin appears to impact the reaction, but it is difficult to assess the possibility of aluminum dissolution from the kaolin, as Al (VI) in the metakaolin can be detected at the same position. According to [55], kaolinite was found to be soluble in lime-based systems, whereas Coudert et al. [7] reported that kaolin remained inert in alkali-activated fly ash mixtures, thus suggesting different behaviors of both systems. The limited reactivity of kaolinite observed may be due to the presence of other highly reactive phases, such as fly ash or metakaolin, in their study.

#### 3.3.2. Effect of Soil Type on Soilmix Strength

The mechanical properties of soilmixes are compared to plain grouts with the same H_2_O/Na_2_O ratio as a function of initial packing density and presented in Figure 10.

The soilmix based on metakaolin grout with H_2_O/Na_2_O ratio 34 developed significant strength, whereas the plain MK34 grout showed negligible strength after 28 days (Figure 10a). This is due to the presence of granular inclusions. The compressive strength of mixtures with ground silica sand and blended silica sand and kaolin reached 0.2 MPa and 0.8 MPa after 28 days, respectively. Binary-based mixtures exhibited higher compressive strength than the metakaolin-based soil mixes, due to their ability to accommodate a higher amount of water as discussed in the previous section (Figure 7). Notably, at an H_2_O/Na_2_O ratio of 28, the binary-based mixtures achieved relatively high strengths, reaching 8 MPa after 28 days (Figure 10b).

Although NMR analyses (Figure 9) reveal that soils such as kaolin and ground silica are largely inert, they have a positive impact on mechanical properties. Soilmixes can be considered composite materials comprising multiple phases, like concrete. Their compressive strength is affected by several factors, including the water-to-binder ratio, compaction level, binder-to-aggregate ratio, and the grading, size, and shape of aggregates [56]. Moreover, kaolinite platelets can act as fillers in alkali-activated fly ash systems, as reported in [7].

Mechanical results obtained on soilmixing mixtures are generally compared to specifications defined for a given construction project based on engineering studies. Target UCS (unconfined compressive strength) specifications are established on a project-by-project basis and requirements for soil improvement. Execution standards do not recommend any specific strength class for suitable deep soil mixing [57]. However, some unconfined compressive strength (UCS) ranges of improved soil have been proposed upon different soil types and binders to assess the effectiveness of deep soil mixing [58]. To achieve significant ground improvement in clayey soil, the UCS performance should fall within the suggested strength ranges (0.2–5 MPa) for deep mixing. This indicates that binary mixtures show potential for soil mixing technology but can be further optimized by balancing the durability offered by the geopolymer gel and the mechanical strength provided by the bound water.

## 4. Conclusions

The potential use of alkali-activated materials and geopolymer for deep soil mixing technique was investigated in this experimental study at different scales. The grouts were first characterized to investigate the influence of the type of precursor on the early-age structuration, phase composition, and engineering properties in the long-term. Then, the influence of H_2_O/Na_2_O ratios on the mechanical performances of grouts and soilmixes was highlighted to reproduce partly the deep soil mixing process.

The activated slag, activated metakaolin, and binary mixtures showed distinct reaction mechanisms. The presence of calcium, introduced by the slag, significantly influenced the advancement of the reaction and strength development, as observed in 75% metakaolin −25% slag mixtures.The analysis of phase composition, chemically bound water, and heat flow profiles highlighted distinctive physicochemical properties of binary metakaolin–slag compared to plain metakaolin or plain slag. These properties resulted from the synergistic interaction between metakaolin and slag.The reaction mechanisms affected the water sensitivity of the materials, resulting in Feret’s relation being inadequate to depict strength evolution as a function of initial porosity in the case of binary and metakaolin grouts designed with H_2_O/Na_2_O ranging from 21 to 34. Slag-based grouts behaved like hydraulic binders, whereas metakaolin-based grouts showed low bound water content and a relatively high sensitivity of precursor reactivity to H_2_O/Na_2_O ratio. Binary mixtures showed the potential to decrease water sensitivity in metakaolin-based grouts. The partial replacement of metakaolin with slag resulted in higher strengths, likely due to reduced porosity (increased bound water) and a potential enhancement in precursor reactivity.The composition and properties of soilmix were evaluated with respect to soil type and water content. Water content can be considered as a major parameter influencing the development of mechanical properties. The inclusion of soils in grouts enhanced mechanical properties due to the granular skeleton. The presence of ground silica sand and kaolin did not significantly impact the physico-chemical evolution or reactivity timescale of the system, except for kaolin, which may partially react over the long term.

Binary soil mixtures composed of 75% metakaolin and 25% slag, with H₂O/Na₂O ratios ranging from 28 to 42, demonstrate potential for soil mixing due to their synergistic reactivity and strength. These mixtures offer durability benefits in challenging soil environments with aggressive waters. However, optimizing the composition, particularly for higher H₂O/Na₂O ratios, is essential to achieve the desired properties of grout–soil mixtures. Evaluating their performance under real soil conditions is crucial for broader development opportunities.

## Figures and Tables

**Figure 1 materials-17-03783-f001:**
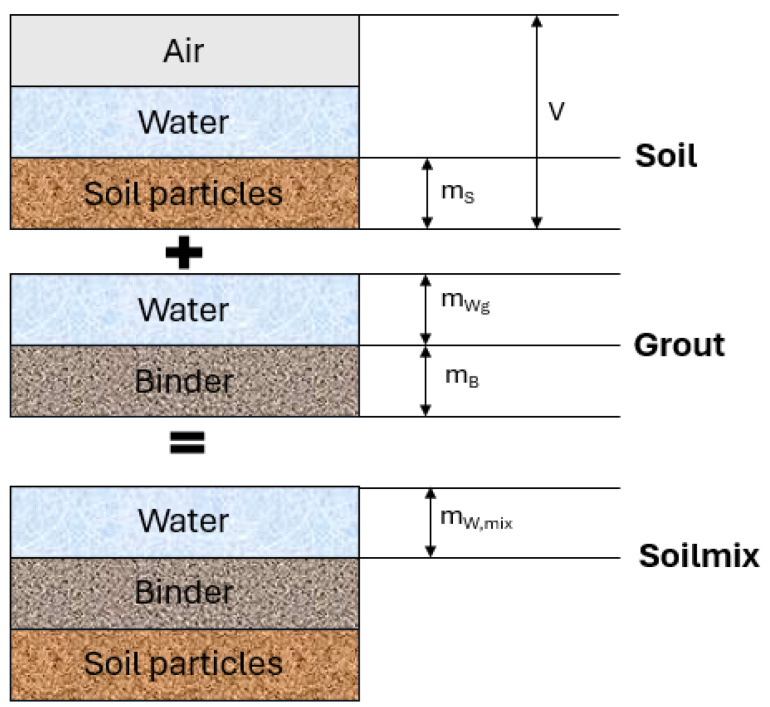
Soilmixing model for the wet process (adapted from [1]). V: total volume of soil, m_S_: mass of soil, m_wg_: mass of water in the grout, m_B_: mass of binder, m_w,mix_: mass of water in the soilmix.

**Figure 2 materials-17-03783-f002:**
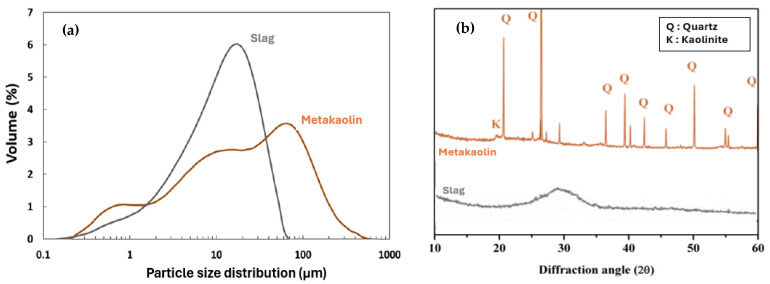
(**a**) Particle size distribution and (**b**) X-ray diffraction analysis of slag and metakaolin.

**Figure 3 materials-17-03783-f003:**
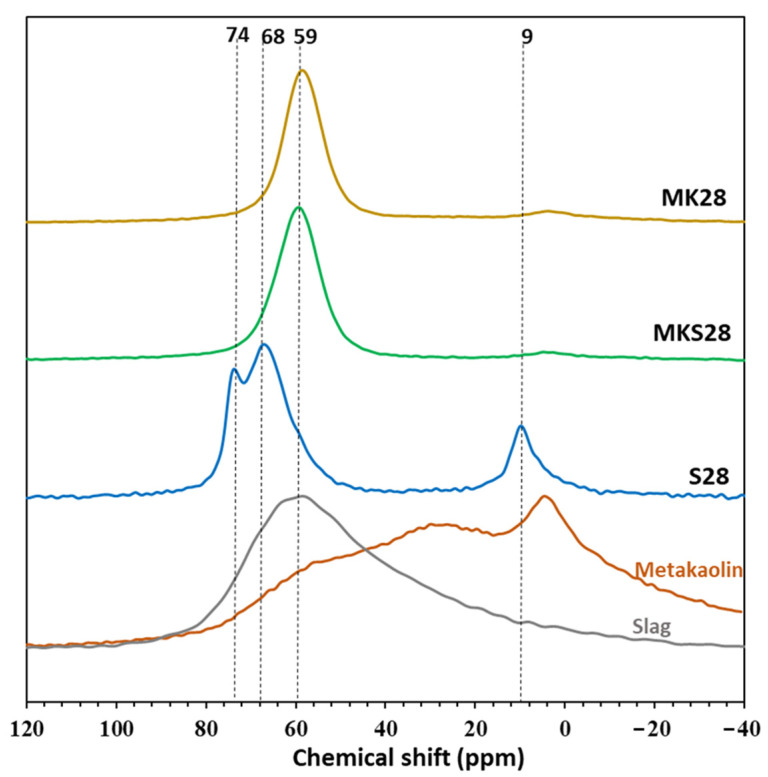
^27^Al MAS NMR of raw metakaolin, raw slag and activated materials after 250 days.

**Figure 4 materials-17-03783-f004:**
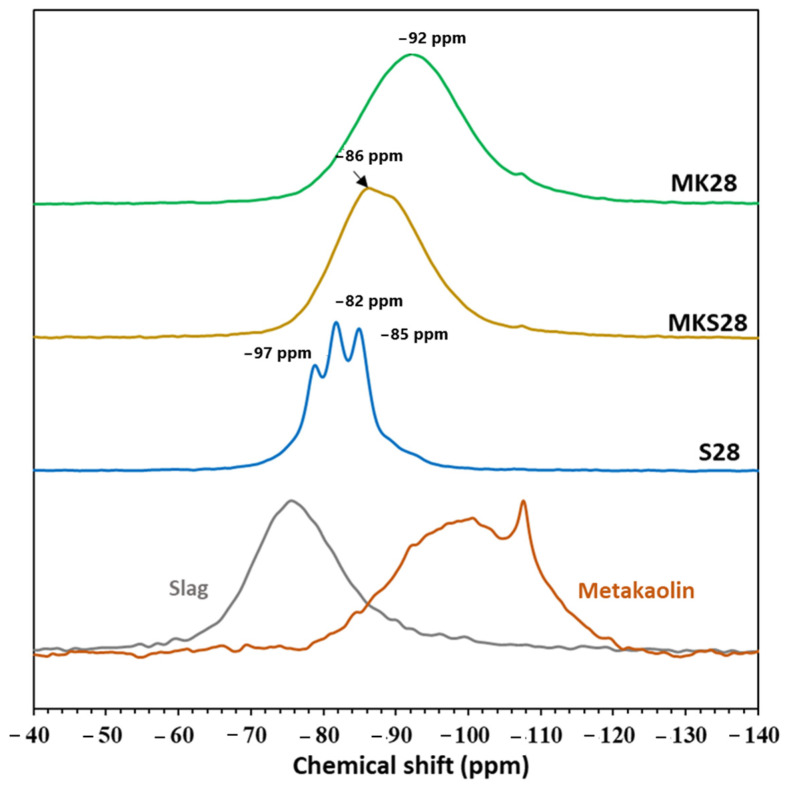
^29^Si MAS NMR of raw metakaolin, raw slag and activated materials after 250 days.

**Figure 5 materials-17-03783-f005:**
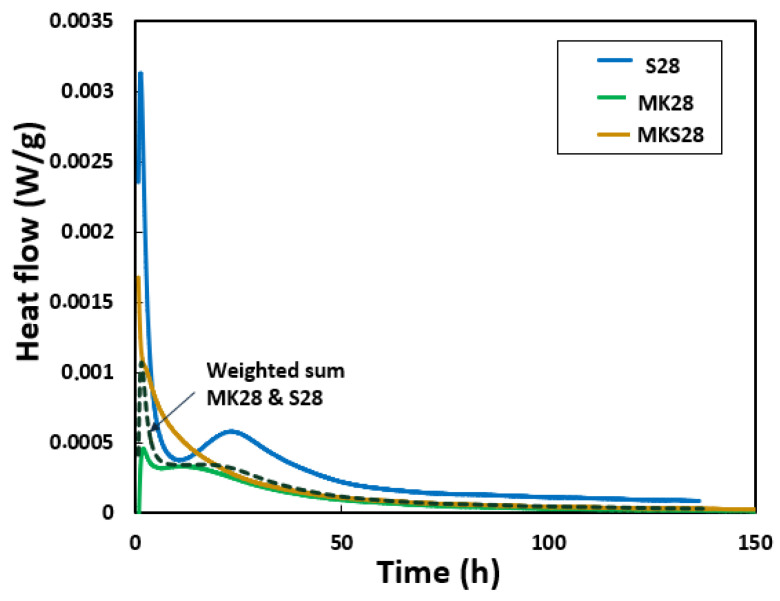
Evolution of heat flow as a function of time.

**Figure 6 materials-17-03783-f006:**
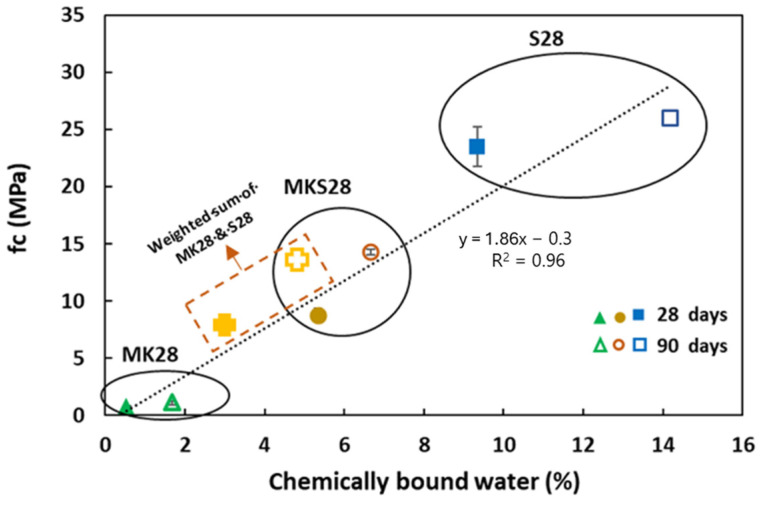
Strength development as a function of chemically bound water.

**Figure 7 materials-17-03783-f007:**
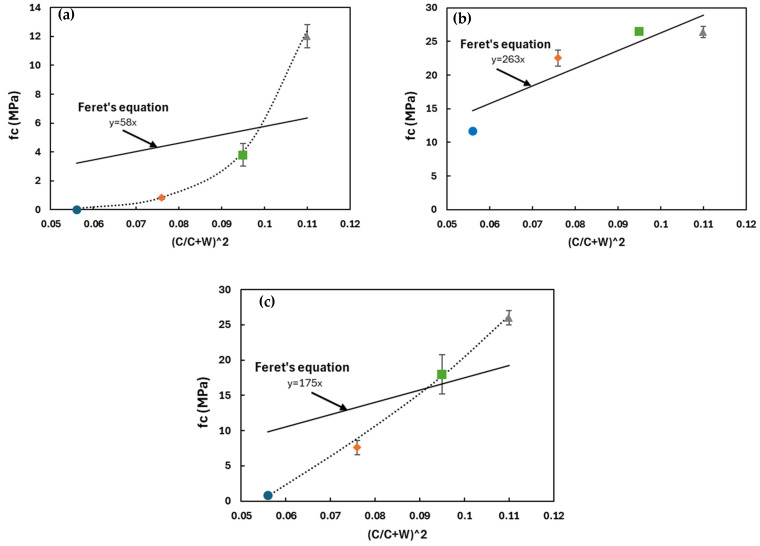
Comparison of experimental compressive strength and values calculated by Feret’s equation at 28 days in (**a**) plain metakaolin (**b**) plain slag and (**c**) binary metakaolin–slag mixtures. Dotted lines correspond to a model to describe the experimentally observed evolution.

**Figure 8 materials-17-03783-f008:**
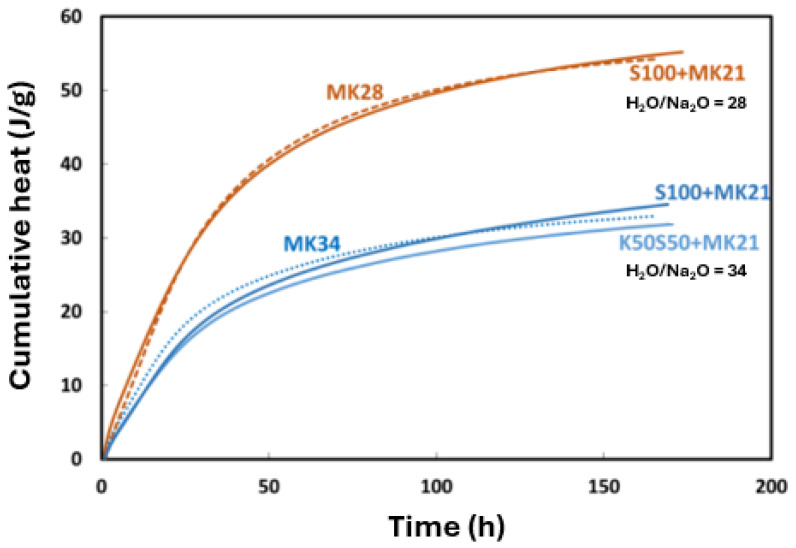
Cumulative heat of soilmixing mixtures per unit mass of grout compared to grouts with H_2_O/Na_2_O ratio of 28 and 34, respectively.

**Figure 9 materials-17-03783-f009:**
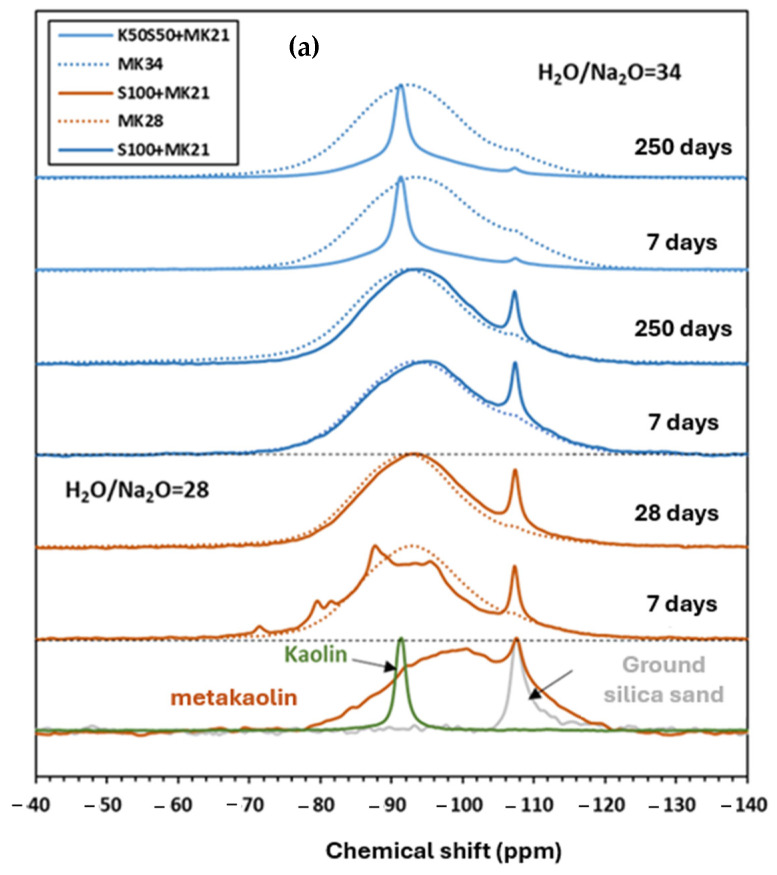
(**a**) ^29^Si MAS NMR and (**b**) ^27^Al MAS NMR spectra of raw materials and soilmix mixtures.

**Figure 10 materials-17-03783-f010:**
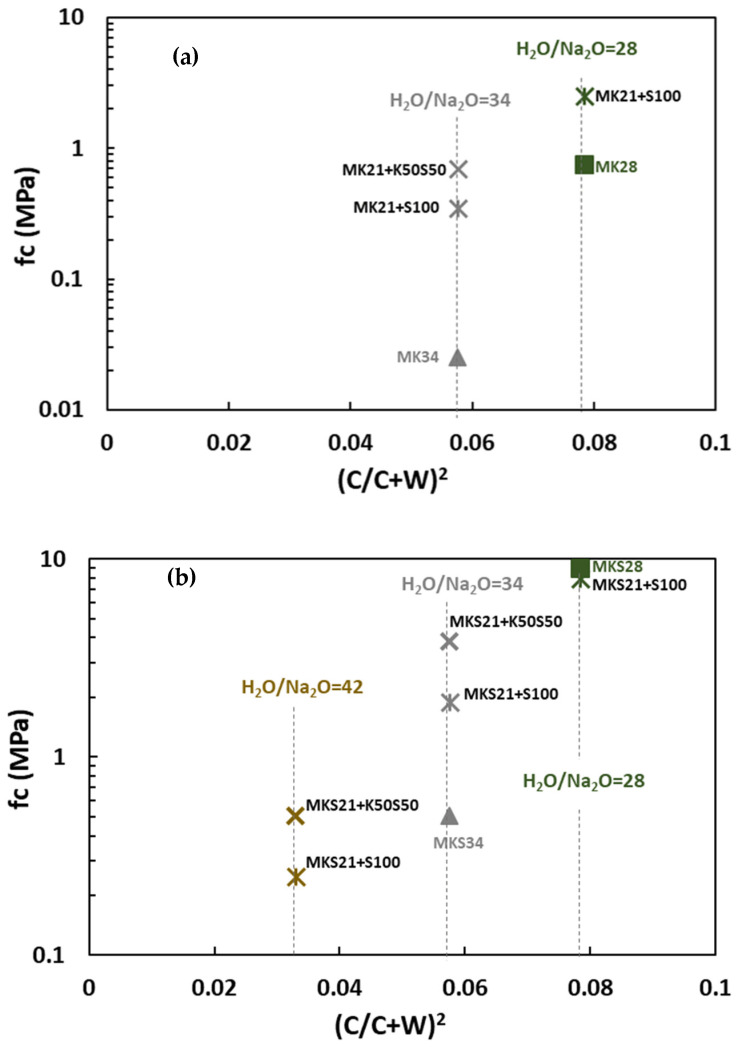
Compressive strength of grouts and soilmixing mixtures for (**a**) metakaolin (**b**) metakaolin and slag-based materials.

**Table 1 materials-17-03783-t001:** Chemical composition (mass fraction. %) and physical properties of metakaolin and slag.

Composition (wt.%)	Metakaolin	Slag
SiO_2_	67.1	37.2
Al_2_O_3_	26.8	10.5
CaO	1.12	43.2
Fe_2_O_3_	2.56	0.6
MgO	0.11	7.0
SO_3_	-	0.1
Cl^−^	-	0.01
TiO_2_	1.3	0.5
Na_2_O	0.01	0.6
Median diameter d_50_ (µm)	18	10
Specific surface BET (m^2^.g^−1^)	16.5	0.45
Density	2.63	2.90

**Table 2 materials-17-03783-t002:** Compositions of the studied mixtures at H_2_O/Na_2_O = 28.

Mixtures			MK28	S28	MKS28
Materials		Density	Compositions (g/L)		
Dry materials	Metakaolin	2.63	604	-	464
	Slag	2.9	-	666	155
	Bentonite	2.5	13	13	13
Solution	Na-silicate	1.55	525	525	525
	NaOH pellets	1.46	2.7	2.7	2.7
	Water	1	429	429	429
Water to solid ratio (W/S wt.%)			0.85	0.79	0.84
Atomic ratio		Si/Al	1.83	4.76	2.2
		Na/Al	1	2.1	1.1
Molar ratio		H_2_O/Na_2_O	28	28	28

**Table 3 materials-17-03783-t003:** Mixtures to study the sensitivity of grouts to water dilution at different H_2_O/Na_2_O ratios.

Materials		Dry Materials			Solution			Water to Solid Ratio W/S (wt.%)	H_2_O/Na_2_O
Composition(g/L)		Metakaolin	Slag	Bentonite	Na-Silicate	NaOH	Water		
Metakaolin-based grouts	MK21	732	-	9	636	3.3	310	0.65	21
	MK24	671	-	11	583	3	367	0.75	24
	MK28	604	-	13	525	2.7	429	0.85	28
	MK34	518	-	15	450	2.3	511	1	34
Slag-based grouts	S38	-	861	14	348	3.5	476	0.65	38
	S44	-	780	16	315	3.2	526	0.75	44
	S50	-	711	17	288	2.9	567	0.85	50
	S62	-	608	19	246	2.5	623	1	62
Binary metakaolin–slag mixtures (75/25 wt.%)	MKS21	562	187	9	636	3.3	310	0.64	21
	MKS24	515	172	11	583	3	367	0.73	24
	MKS28	464	155	13	525	2.7	429	0.84	28
	MKS34	397	133	15	450	2.3	511	1	34

**Table 4 materials-17-03783-t004:** Composition of soilmixing mixtures made from plain metakaolin grout MK21.

Materials		Dry Materials			Solution			Water to Solid Ratio W/S (wt.%)	H_2_O/Na_2_O
Composition g		Metakaolin	Slag	Bentonite	Na-Silicate	NaOH	Water		
Metakaolin-based grouts (1 L)	MK21	732	-	9	636	3.3	310	0.65	21
Model soil (1 L)		Ground silica sand		Kaolin		Water		Ratios of soil mixtures	
	S100	2102		-		207		0.85	28
	S100	1618		-		390		1	34
	S50K50	749		749		390		1	34

## Data Availability

The original contributions presented in the study are included in the article, further inquiries can be directed to the corresponding author.

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
