# Peer review of "Design of Alkali-Activated Materials and Geopolymer for Deep Soilmixing: Interactions with Model Soils"

_materials, 2024, doi:10.3390/ma17153783_

Round 1

Reviewer 1 Report

Comments and Suggestions for Authors

The subject of this article „Design of alkali-activated materials and geopolymer for deep 2 soilmixing: interactions with model soils“ is interesting and usefull for mining industry.

is interesting, but  the results interpretation is unclear. The authors focuses on the use of alkali-activated materials and geopolymers grouts in deep  soilmixing. Three types of grouts, incorporating metakaolin and/or slag, were characterized at different levels to understand the development of their local structure and macroscopic properties. Feret’s approach was used to understand the development of compressive strength at different water-to-solid ratio. The performance of the soilmix was assessed by using combinations of the investigated materials and model soils with different clay contents. The results suggested that incorporating calcium reduced the water sensitivity of the materials, which is an important parameter in soilmixing. The addition of soils to grouts resulted in improved mechanical properties, attributed to the influence of the granular skeleton. Their presence did not alter the physico-chemical evolution  of the system or affect the reactivity timescale, except for the kaolin, which can  partially react over the long term.

Although the subject of this article is interesting, but  the results interpretation is unclearand need additional explanation. Some comments are as follows. I hope they will be helpful for the authors to improve the manuscript in the future.

Page 8, line 1-2. The sentence „  It should be noted that the curve obtained from the weighted sum of MK28 and S28 250 heat flows does not correspond to the profile of the flow obtained for MKS28“ needs to be clarified. How was calculated  weighted sum of MK28 and S28 250 ?  

Page 8, Fig 9. The label needs to be changed from French to English .

Page 9. Need to be corrected    (Error! Reference source not 301 found.)“

Page 10. „ In  the case of slag-based grouts, C-A-S-H were formed, and a significant amount of water  was thus chemically bound, which decreases the porosity, whereas in the case of metakaolin, most of water was still free water in the pores as previously mentioned (Figure 5)“. The study does not provide enough evidence to state that „a significant amount of water  was thus chemically bound, which decreases the porosity“ . No porosity studies are provided.

Page 12. Fig 7(a,b). The label needs to be changed from French to English. The composition indicated in the figure is not presented in Table 3.

Please clarify in the conclusions what is „simple weighted activity of 440 the metakaolin and slag-based grouts individually“. This must be explained in the materials and methodology section.

Author Response

The authors would like to thank the reviewers for their efforts in handling this scientific work. The interesting remarks are gratefully acknowledged and will improve the quality of this paper. All comments and suggestions have been addressed in the new version of the manuscript. The modifications are marked in red characters in the revised manuscript to facilitate the review process. The attached file presents the changes performed and responses to the reviewer’s comments and recommendations, in the order they were raised.

Reviewer 2 Report

Comments and Suggestions for Authors

The authors produced alkali-activated materials and geopolymeric mortars, with three types of grout, incorporating metakaolin and/or slag. The manuscript is interesting and has merit for publication after some revisions:

>Abstract. Please add the ideal composition into the mix that optimizes results. Finalize the abstract on the potential for the construction sector, etc.;

>Page 3. Line 94-109. Please address the economic aspect of using metakaolin, for example, it can contribute to reducing the final cost, there is the possibility of reusing waste, etc.;

> Regarding the raw material, why did the authors not present x-ray diffraction (XRD), laser particle size distribution, specific mass and specific surface area Blaine? How was the granulometry control?

>Page 6. “These wide peaks indicate the amorphous nature of the metakaolin”. Actors could add references relating the indicated sign to the degree of perfection (crystalline/amorphous);

> “The peak at around 9 ppm is related to Al(VI) in hydrotalcite, which is generally considered a poorly…………” It is interesting to add the XRD results, this way it is better to understand the results;

>Figure 3. Please discuss further the behavior of sample S28, with three signals and shifted;

> Authors must compare the mechanical results obtained whether they are suitable for practical applications, as recommended by standards;

> Why didn't the authors perform an SEM analysis of morphology and EDS mapping to understand the distribution of the components?

> Conclusion. The authors only addressed experimental results. It did not indicate the advantage of the systems produced, compared to the most used ones;

Comments on the Quality of English Language

Minor editing of English language required

Author Response

(The authors gave the same response as above.)

Reviewer 3 Report

Comments and Suggestions for Authors

The following parts of this article need to be revised

1. In lines 154 and 155, the author mentioned S100 and K50S50. However, the author did not give a detailed mix ratio. In order to improve the completeness of the article, detailed mix ratios need to be given.

2. The mix ratios in Table 3 are all expressed in relative proportion format. In order to enhance the repeatability of the experiment, I suggest a format similar to Table 2, using the material composition per liter.

3. In Table 2, for different mix ratios, the molar ratio of water and sodium oxide is 28. Why choose this value? Is there any engineering meaning?

4. In Table 3, for the mixture of kaolin and slag, their mass ratio is 75% and 25%. The author needs to give a reason for using this ratio.

5. The author conducted macroscopic experiments such as strength and microscopic experiments such as hydration heat and NMR. Generally speaking, when introducing experimental results, the macroscopic experimental results should be explained first, and then the microscopic experimental results.

6. Figure 4 shows the hydration heat release rate, but it is only 50 hours. In the previous experimental part, the author said in line 196 that the hydration heat was measured for 7 days. I suggest that the author plot the complete hydration heat results. In addition to the hydration heat release rate, the cumulative hydration heat results must also be given.

7. In Figure 5, the author shows the relationship between chemically bound water and strength. In addition to chemically bound water, the cumulative hydration heat may also have some relationship with strength. I suggest that the author try whether there is an obvious correlation between strength and cumulative hydration heat.

8. There is a problem with the reference in line 301.

9. In Figure 6, the author tried to regress the strength, but the test points were too few, so few test points will cause regression errors and affect the conclusions of this article.

10. Regarding the regression of strength, Figure 8 also has the same problem.

11. The English in this article is difficult to understand and needs to be revised by a specialized agency. Some words may not be English, I guess they are French. Both the words in the figure and the words in the text need to be checked.

Comments on the Quality of English Language

English very difficult to understand/incomprehensible.

Proof reading is necessary.

Author Response

(The authors gave the same response as above.)
